# On-Chip Integration of Pressure Plus 2-Axis (X/Z) Acceleration Composite TPMS Sensors with a Single-Sided Bulk-Micromachining Technique

**DOI:** 10.3390/mi10070473

**Published:** 2019-07-15

**Authors:** Jiachou Wang, Fang Song

**Affiliations:** 1State Key Lab of Transducer Technology, Shanghai Institute of Microsystem and Information Technology, Chinese academy of Sciences, Shanghai 200050, China; 2Center of Materials Science and Optoelectronics Engineering, University of Chinese Academy of Sciences, Beijing 100049, China; 3College of Mechanical Engineering, Shanghai University of Engineering Science, Shanghai 201620, China

**Keywords:** two-axis (X/Z) accelerometer, pressure sensor, tire pressure monitoring system (TPMS), on-chip integration, single-side bulk-micromachining process

## Abstract

A novel on-chip integration of pressure plus 2-axis (X/Z) acceleration composite sensors for upgraded production of automobile tire pressure monitoring system (TPMS) is proposed, developed, and characterized. Herein, the X-axis accelerometer is with the cantilever beam-mass structure and is used for automatically identifying and positioning each of the four wheels. The IC-Foundry-Compatible low-cost batch fabrication technique of MIS (i.e., Micro-openings Inter-etch and Sealing) is employed to only fabricate the device from the front side of (111) silicon wafer, without double-sided micromachining, wafer bonding, complex Cavity-SOI (Silicon on Insulator) processing, and expensive SOI-wafer needed. Benefited from the single-wafer front-side fabrication technique on ordinary single-polished wafers, the fabricated composite TPMS sensor has the advantages of a small chip-size of 1.9 mm × 1.9 mm, low cross-talk interference, low-cost, and compatible process with IC-foundries. The fabricated pressure sensors, X-axis accelerometer and Z-axis accelerometer, show linear sensing outputs, with the sensitivities as about 0.102 mV/kPa, 0.132 mV/kPa, and 0.136 mV/kPa, respectively. Fabricated with the low-cost front-side MIS process, the fabricated composite TPMS sensors are promising in automotive electronics and volume production.

## 1. Introduction

Silicon-based pressure sensor and accelerometer have been intensively developed and widely used in industry and costumer application due to their tiny size, low cost, and linear output property [1]. Recently, along with the market expansion of automobile electronic, great technical efforts have been motivated to develop pressure plus Z-axis acceleration composite tire pressure monitoring system (TPMS) sensor, where the Z-axis accelerometer is mainly used to detect the wheel-rotation-induced centrifugal acceleration for waking up the pressure sensor to monitor tire-pressure in due time. With the rapid development of science and technology and the improvement of people’s living standards, today, for further improving vehicle safety and comfort, an additional X-axis accelerometer is proposed and integrated into the composite TPMS sensor for automatic identifying and positioning each of the four wheels, even if the wheel positions are mutually exchanged in the 4S shops. When the car experiences the first turning, the different behaviors of the four wheels in the X-axis acceleration will be detected and recorded in the Microcontroller Unit (MCU) processor [2,3]. Now the worry lies in that the users would like to enjoy the improved function, but do not want to pay more for it, thereby the only way to avoid the additional cost and reduce the device size is to realizing monolithic integration of the three sensors in a cheap non-SOI (Silicon on Insulator) wafer. Additionally, for compatible manufacturing in standard IC-foundries, developing a single-sided process should be highly demanded. 

To date, many reported composite TPMS sensors have consisted of only a pressure sensor and a Z-axis accelerometer, and formed by the hybrid packaging scheme or the on-chip integrating scheme. Compared with the hybrid packaging scheme for realizing simultaneous detection of pressure and acceleration, the latter scheme seems to be a good choice due to its advantages of smaller size, lower cost, higher reliability, and higher fabrication throughput. Thus far, various approaches, such as surface-micromachining [4,5,6], double-side bulk-micromachining [7,8], Cavity-SOI process [9], double-side CMOS post process [10], and MEMS-CMOS bonding process [11] have been developed and used to fabricate the on-chip integrating multifunctional sensor. Unfortunately, the fabricated sensors have more or less shortcomings [1], such as low sensitivity, large device-size, complex fabrication process, and high fabrication cost. More important is that, unlike the integration of X- and Y-axis accelerometers laid in-plane, integration of vertical Z- and horizontal X-axis accelerometers is quite challenging.

To solve the aforementioned problems, this study proposes and develops an IC-Foundry-Compatible low-cost batch fabrication technique to realize monolithic integration of pressure plus two-axis (X/Z) acceleration sensors in cheap non-SOI silicon wafers. The design, fabrication, and the characterization of the composite TPMS sensor will be detailed in the following sections.

## 2. Design

The three-dimensional schematic of our proposed monolithic composite TPMS sensor structure, shown in Figure 1, consists of a Z-axis accelerometer with dual-cantilever-mass configuration, an X-axis accelerometer with cantilever beam-mass configuration, and a pressure-sensor located inside a *PS*^3^ structure for packaging-stress release [12]. For the design of the Z-axis accelerometer without plane sensing direction, the movable gap between the seismic mass, and the silicon substrate is embedded into the silicon substrate rather than being formed by silicon–silicon (or silicon-glass) bonding, which can be controlled by fabrication process to optimize the squeeze-film damping and overload protection capability. The X-axis accelerometer with an in-plane sensing direction is laid out along <211> orientation and surrounded by the self-stop protection gap whose dimension is dependent on the desired acceleration range. As is shown in Figure 1, all of these different sensing components are all only fabricated from the front side of ordinally (111) silicon wafer, without double-sided micromachining, wafer bonding, Cavity-SOI, and expensive SOI-wafer was needed.

### 2.1. Pressure Sensor

For the design of the pressure sensor, two rows of micro-trenches are opened along the <211> orientation to construct the single-crystal silicon hexagonal-shaped pressure sensing diaphragm by laterally excavating the silicon along <211> and <110> orientation, where the trench size is 4 μm × 4 μm and the inter-row separation is 48 μm. As is depicted in Figure 2, the length and the width of the hexagonal shaped diaphragm, *l*, *w*, are designed as 350 μm and 56 μm, respectively. The thickness of the diaphragm is 8 μm. The reference-pressure cavity beneath the diaphragm is embedded into the *PS*^3^ structure [12]. The pressure-induced stress distribution on the diaphragm is simulated by ANSYS software, and the Wheatstone-bridge piezoresistors are laid out at the locations with maximum piezoresistive sensitivity shown in Figure 2. The ANSYS-simulated output voltage distribution on the upper surface of the diaphragm is shown in Figure 3, where the input pressure is set as 450 kPa, and the applied voltage on the Wheatstone-bridge circuit is set as 3.3 V. As shown in Figure 3, the theorical output voltage of the pressure sensor is about 47 mV and the sensitivity is 0.104 mV/kPa. 

### 2.2. Z-Axis Accelerometer

As is depicted in Figure 1, the Z-axis accelerometer is with the dual-cantilever-mass configuration. According to the analysis in Ref. [13], the deflection along the sensing direction of the cantilever (i.e., the normal direction to the (111) wafer) can be expressed as
(1)$$w_{1}\left(x\right)=\frac{a_{2}b_{2}h_{2}\rho a\left(3a_{1}+1.5a_{2}-x\right)x^{2}}{Eb_{1}h_{1}^{3}}$$
where *a* is the externally applied acceleration, *a*_1_, *b*_1_, and *h*_1_ are the cantilever length, width, and thickness, respectively; *a*_2_, *b*_2_, and *h*_2_ are the seismic-mass length, width, and thickness, respectively; *E* is Young’s modulus of silicon, *ρ* is the density of single-crystal silicon. As the seismic-mass is much thicker than the cantilever, i.e., *h*_2_ >> *h*_1_, the deflection of the mass is negligible. Therefore, the deflection of the seismic-mass can be expressed as
(2)$$w_{2}\left(x\right)=w_{1}\left(a_{1}\right)+w_{1}^{\prime }\left(a_{1}\right)\left(x-a_{1}\right)$$

At a location of *x* = *a*_1_ + *a*_2_, therefore, the maximum displacement, *w*_max_, can be expressed as
(3)$$w_{\max}=\frac{a_{1}a_{2}b_{2}h_{2}\rho a\left(2a_{1}+4.5a_{1}a_{2}+3a_{2}^{2}\right)}{Eb_{1}h_{1}^{3}}$$

The generated stress on the top-surface of the cantilever can be expressed as
(4)$$T\left(x\right)=\frac{3a_{2}b_{2}h_{2}\rho a\left(a_{1}+0.5a_{2}-x\right)}{b_{1}h_{1}^{2}}$$

According to Equation (4), the maximum stress for the piezoresistance is located near the root of the cantilever, the designed piezoresistor pattern is shown in Figure 1. For the dual-cantilever-mass accelerometer with a half-bridge circuit consisting of p-type piezoresistors, the half-bridge circuit is shown in Figure 4 and the normal sensitivity can be analyzed as
(5)$$S=\frac{3a_{2}b_{2}h_{2}\rho \pi_{44}V_{in}}{4b_{1}h_{1}^{2}l_{p}}\int_{0}^{l_{p}}\left(a_{1}+0.5a_{2}-x\right)dx=\frac{3a_{2}b_{2}h_{2}\rho \pi_{44}V_{\mathrm{in}}}{4b_{1}h_{1}^{2}l_{p}}\left(a_{1}+a_{2}-0.5l_{p}\right)$$
where *π*_44_ is the piezoresistive coefficient component [13], *V*_in_ is the supplied voltage on the half-bridge circuit and *l_p_* is the piezoresistor length.

Similarly, for an acceleration in the sensor chip plane and vertical to the cantilever direction, the maximum stresses on the two edges of the cantilever can be calculated as
(6)$$T_{\mathrm{trans}}=\frac{3a_{2}b_{2}^{2}h_{2}\rho a\left(a_{1}+0.5a_{2}\right)}{4b_{1}h_{1}\left(b_{1}^{2}-1.5b_{1}b_{2}+0.75b_{2}^{2}\right)}$$

Therefore, according to Equations (4) and (5), lateral sensitivity to acceleration vertical to the cantilever is:(7)$$S_{\mathrm{trans}}=\frac{b_{2}h_{1}S}{4\left(b_{1}^{2}-1.5b_{1}b_{2}+0.75b_{2}^{2}\right)}$$

In our design of the Z-axis accelerometer, the cantilever length, width and thickness are 120 μm, 50 μm and 7 μm, respectively. The seismic-mass length, width and thickness are 700 μm, 600 μm and 38 μm, and the piezoresistor length is 24 μm. For a 3.3 V power supplied on the half-bridge circuit, the normal sensitivity is designed as 0.135 mV/g. Additionally, the lateral acceleration induced sensitivity is about only 0.46% as large as that caused by the normal acceleration of the same value. Therefore, the lateral effect caused by a X/Y-axis acceleration is negligible.

### 2.3. X-Axis Accelerometer

Please refer the X-axis accelerometer configuration schematically shown in Figure 1, the displacement, *w*_d_(*x*), can be calculated as [13]
(8)$$w_{d}\left(x\right)=\frac{2l_{2}w_{2}\rho a\left(3l_{1}+1.5l_{2}-x\right)x^{2}}{Ew_{1}^{3}}$$
where *l*_1_, *w*_1_, and *t*_1_ are the cantilever length, width, and thickness, respectively; *l*_2_, *w*_2_, and *t*_2_ are the seismic-mass length, width, and thickness, respectively. Since the seismic mass is much thicker than the cantilever, deflection of the mass is reasonably negligible. Therefore, the maximal deflection *w*_max_ occurs at the far end of the mass (i.e., *x* = *l*_1_ + *l*_2_), with the value as
(9)$$w_{\max}\left(l_{1}+l_{2}\right)=\frac{2l_{1}l_{2}w_{2}\rho a\left(2l_{1}^{2}+4.5l_{1}l_{2}+3l_{2}^{2}\right)}{Ew_{1}^{3}}$$

The generated stress on the surface of the cantilever can be expressed as
(10)$$T\left(x\right)=Ew^{\prime }w_{d}^{\prime \prime }\left(x\right)=\frac{6l_{2}w_{2}w^{\prime }\rho a\left(2l_{1}+l_{2}-2x\right)}{w_{1}^{3}}$$
where *w*´ is the distance between the surface of the cantilever and the neutral plane (i.e., the middle plane) of the cantilever. Considering the lithography alignment accuracy and the stress distribution, two piezoresistors (*R*_Canti.1_ and *R*_Canti.2_) are symmetrically located on the surface, where $w^{\prime }=w_{1}/4$, and near the root of the cantilever. Together with the other two reference piezoresistances (*R*_Ref_), the four piezoresistors form a half-bridge circuit that is shown in Figure 5. Therefore, the X-axis accelerometer output voltage, *V*_out_, is calculated as
(11)$$V_{\mathrm{out}}=\frac{3\pi_{44}V_{\mathrm{in}}l_{2}w_{2}\rho a}{8w_{1}^{2}l_{p}}\int_{0}^{l_{p}}(2l_{1}+l_{2}-2x)dx=\;\frac{3\pi_{44}V_{\mathrm{in}}l_{2}w_{2}\rho a}{8w_{1}^{2}}\left(2l_{1}+l_{2}-l_{p}\right)$$

The sensitivity can be analyzed as
(12)$$S=\frac{3\pi_{44}V_{\mathrm{in}}l_{2}w_{2}\rho }{8w_{1}^{2}}\left(2l_{1}+l_{2}-l_{p}\right)$$

Additionally, according to the piezoresistors arrangement in Figure 1 and half-bridge circuit in Figure 5, *R*_Canti.1_ and *R*_Canti.2_ will increase or decrease their resistance value simultaneously when an acceleration is vertical to the sensor chip plane and applied on the X-axis accelerometer, the output voltage, *V*_out1_, is equal null. Therefore, the normal effect caused by a Z-axis acceleration is negligible for the design of the X-axis accelerometer.

In our design of the X-axis accelerometer, the cantilever length, width, and thickness are 300 μm, 13.5 μm, and 38 μm, respectively. The seismic-mass length, width, and thickness are 1100 μm, 450 μm, and 38 μm, and the piezoresistor length is 46 μm, respectively. For a 3.3 V power supplied on the half-bridge circuit, the sensitivity is designed as 0.104 mV/g.

## 3. Fabrication

The proposed single-side bulk-micromachined composite TPMS sensors are formed in an ordinary single-polished (111) silicon wafers. The cross-sectional fabrication steps are shown in Figure 6, where the cross-section is cut along the broken-line of D-D´ in Figure 1. The steps of (a)–(b) are mainly used to fabricate the piezoresistors, the pressure sensitive diaphragm and the vacuum cavity and detailed fabrication flow can refer to Ref. [14]. The following steps of (c)–(g) are employed for fabricating the X- and Z-axis accelerometers and step-by-step described as follows:(c)Two parallel trenches along the <211> -orientation are patterned and etched by using Deep-RIE (reactive ion etching) to a depth of 8 μm to define the shape and thickness of the dual-cantilever of the Z-axis accelerometer. Thereafter, 0.4 μm-thick TEOS (Tetraethyl orthosilicate) film is LPCVD (low pressure chemical vapor deposition) deposited to protect the vertical sidewall surface of the trenches from the following anisotropic wet etching. Then, the TEOS film at the trench bottom is removed by using RIE to expose bare silicon while the TEOS film on the trench sidewalls is still remained. Next, Deep-RIE is used again to deepen the trenches to form the sacrificial gaps for subsequent lateral under-etch. The depth of the sacrificial gap should be larger than that of the seismic-mass plus its movable gap.(d)According to the (111) silicon anisotropic etching properties, the wafer is dipped into 80 °C 25% TMAH (Tetramethyl ammonium hydroxide) etchant to laterally under-etch through the bare-silicon sacrificial gaps along <110> orientation to release the dual-cantilever structure.(e)With photoresist as an etching mask, even deeper trenches etching is processed by Deep-RIE to define the shape of the beam and mass of the X-axis accelerometer, the mass of the Z-axis accelerometer, as well as the cantilever-like *PS*^3^ structure. Then, 0.4 μm-thick TEOS film is deposited to cover the trench-sidewall. Then, the passivation film at the trench-bottom is stripped to expose bare-silicon at the bottom. Deep-RIE trench deepening is processed again to form a bare-silicon bottom trench-segment for following lateral wet-etch, where the bottom-segment depth equals the seismic-mass movable gap of the Z-axis accelerometer.(f)Aqueous TMAH is employed again to laterally under-etch from the exposed bare-silicon bottom segment until the cantilever-like *PS*^3^ structure, the spring and mass of the X-axis accelerometer, and the mass of the Z-axis accelerometer are all released into free standing.(g)Electric contact holes are opened, and aluminum thin-film is sputtered, patterned, and sintered for piezoresistors interconnection. Finally, Deep-RIE is used to fully release the two accelerometers from the front-side of the silicon wafer.

Figure 7a shows the SEM image of the fabricated on-chip integration of pressure plus X-/Z-axis acceleration composite TPMS sensor by implementing the fabrication process, as illustrated in Figure 6, where the chip size is as small as 1.9 mm × 1.9 mm × 0.45 mm. Figure 7b shows the close-up view of the X-axis accelerometer, whose thickness is identical to the thickness of the seismic-mass of Z-axis accelerometer and the *PS*^3^ structure. The pressure sensor in Figure 7c is surrounded with the *PS*^3^ structure, and an embedded vacuum-cavity can be clearly observed when it manually broke out the pressure-sensing diaphragm. In order to clearly observe the Z-axis accelerometer seismic-mass embedded into the silicon substrate, the dual-cantilever-mass structure is broken manually and shown in Figure 7d. 

## 4. Characterization and Discussion

A precise centrifugal-acceleration generation/testing system (DL SY30-3, DongLing Intelligent Vibration and Control Co., Ltd, Suzhou, China) is used to evaluate the performance of the X/Z-axis accelerometer. Figure 8 schematically shows the measurement setup. 3.3 V DC power (MOTECH, LPS305, Suzhou, China) is supplied to the half-bridge circuit of the accelerometer and the output signal of the accelerometer is directly fed into a digital multimeter (Agilent 34401A, Santa Clara, CA, USA) for readout. 

Without any amplification, the Z-axis accelerometer output voltage versus applied Z-axis acceleration is shown in Figure 9, within an acceleration range of 0–120 g, the sensitivity is tested at about 0.136 mV/g under 3.3 V DC supply, which agrees well with the design value of 0.135 mV/g. By using the best fitting-line method [13], the nonlinearity error is measured as about ±0.12% FS. The Z-axis acceleration induced pressure sensor cross-sensitivity is only 0.5 μV/g, which also demonstrates that the effect of the Z-axis acceleration load on the pressure signal is reasonably neglected. As is depicted in Figure 5, when the Z-axis acceleration is applied on the X-axis accelerometer, the two piezoresistors, *R*_canti.1_ and *R*_canti.2_, exhibit the same resistance change, Δ*R*, therefore the X-axis accelerometer has no signal output.

The zero-point offset and full-scale sensing output of the Z-axis accelerometer as functions of temperatures are tested with the results shown in Figure 10. Within the range of −40 °C to 90 °C that is required from the TPMS application, the TCO and the TCS are tested at about 0.25%/°C FS and −0.31%/°C FS for the whole 120 g range, respectively.

Figure 11 shows the X-axis accelerometer output voltage versus applied X-axis acceleration. within the 120 g measured range, the sensitivity is measured as 0.132 mV/g, which is slightly higher than the designed 0.104 mV/g. The discrepancy in sensitivity is probably due to the fabrication tolerance. For example, the fabricated cantilever width is about 13.3 μm, which is slightly thinner than that of the designed cantilever width, 13.5 μm. According to Equation (10), the thinner the cantilever the higher the sensitivity. Based on the results in Figure 10, the nonlinearity is better than ± 0.61% FS. In addition, the X-axis input-acceleration induced Z-axis accelerometer cross-sensitivity is merely 0.846 μV/g. Compared to the Z-axis accelerometer sensitivity of 0.136 mV/g, the crosstalk of the X-axis acceleration to the Z-axis accelerometer is negligible. As shown in Figure 12, within the range of −40 °C to 90 °C, the TCO and TCS are 0.23%/°C FS and −0.21%/°C FS, respectively.

Druck DPI-104 pressure gauge and PV211 handheld pump system are used to test the performance of the pressure sensor integrated in the TPMS sensor. At a room temperature of 20 °C, the measured output voltage versus applied pressure is shown in Figure 13, the sensitivity is 0.102 mV/kPa under 3.3V DC power apply for the Wheatstone bridge, and the nonlinearity is better than ± 0.20% FS. The full measured range of absolute pressure is 700 kPa. The measured sensitivity agrees well with the designed 0.104 mV/kPa. As is shown in Figure 13, the output of the Z-axis accelerometer is recorded simultaneously. The input-pressure induced Z-axis accelerometer crosstalk output is tested as low as 0.120 μV/kPa (i.e., 0.88 mg/kPa), which is small enough to prevent a false trigger. The measurement results also demonstrate that the pressure load does not influence the acceleration signal significantly. Figure 14 shows the zero-point offset drift and full-scale sensing output of the pressure sensor in terms of temperature change from −40 °C to 125 °C. Based on the testing results in Figure 14, the TCO is as low as −0.05%/°C FS, where the full measure range is 700 kPa. Then, the TCS is measured as −0.17%/°C FS.

## 5. Conclusions

In this paper, a novel on-chip integration of pressure plus two axis (X/Z) acceleration TPMS composite sensor has been proposed, designed, fabricated, and characterized. The IC-foundry-compatible fabrication process is implemented only from the front-side of an ordinary (111) silicon wafer. Benefited from the single-side bulk-micromachining technique and optimization design of structure parameter, the composite sensor has the advantages of small chip size of 1.9 mm × 1.9 mm, low cross-talk interference, and low-cost high-throughput IC-foundry batch fabrication capability. The 700-kPa-ranged pressure-sensor sensitivity is measured as 0.102 mV/kPa, with the nonlinearity as ± 0.21% FS. The performance of the 120-g-ranged two-axis accelerometer are measured, exhibiting a sensitivity of 0.132 mV/g, nonlinearity of ± 0.18% FS and the noise floor better than 0.2 g for X-axis accelerometer, and a sensitivity of 0.136 mV/g, nonlinearity as ± 0.12% FS and the noise floor better than 0.2 g for Z-axis accelerometer, respectively. Testing results verified that the developed sensor not only has high performance, but also meet the requirements of upgraded TPMS application. 

## Figures and Tables

**Figure 1 micromachines-10-00473-f001:**
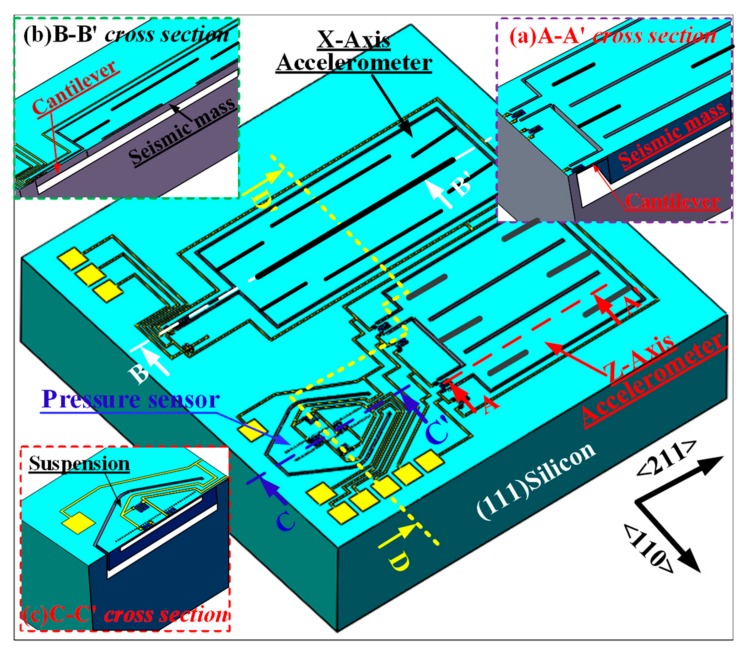
3D sketch of the on-chip integration of pressure plus two-axis acceleration composite tire pressure monitoring system (TPMS) sensor.

**Figure 2 micromachines-10-00473-f002:**
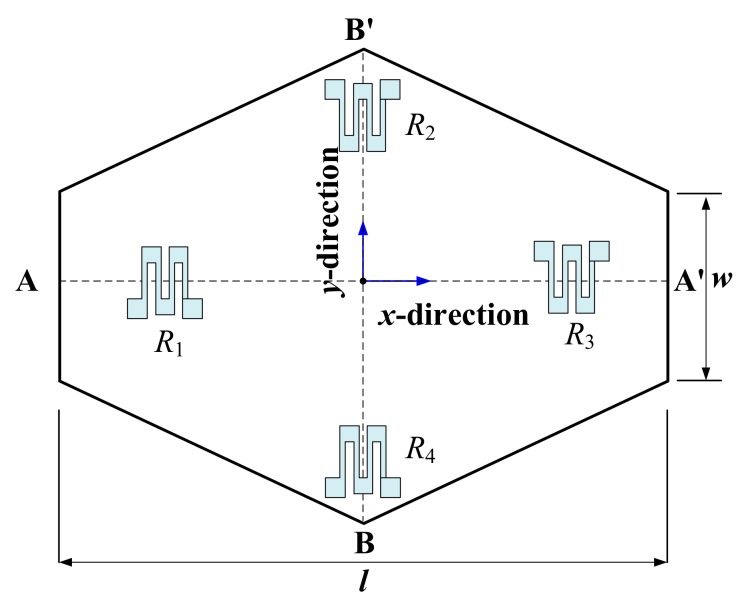
Optimally designed piezoresistor distribution on the pressure sensing diaphragm.

**Figure 3 micromachines-10-00473-f003:**
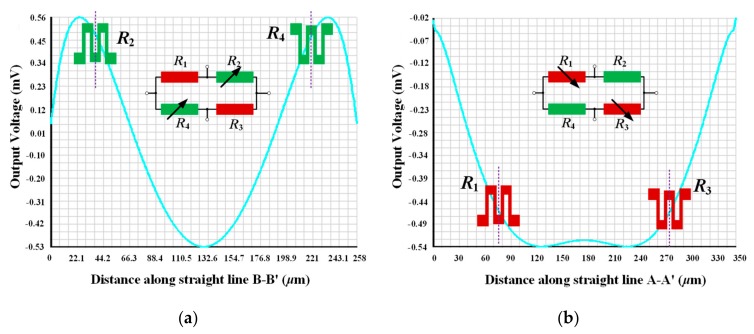
ANSYS-simulated output voltage distribution of the pressure sensor. (**a**) Output voltage distribution along line A-A´ in Figure 2; (**b**) output voltage distribution along line B-B´ in Figure 2.

**Figure 4 micromachines-10-00473-f004:**
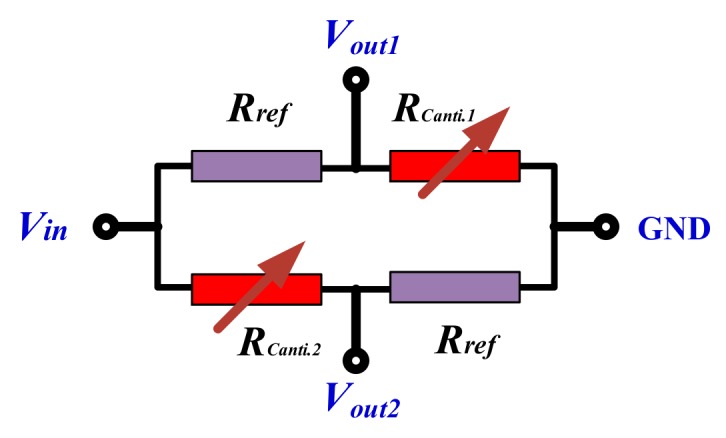
Schematic of the piezoresistive half-bridge circuit for Z-axis acceleration sensing.

**Figure 5 micromachines-10-00473-f005:**
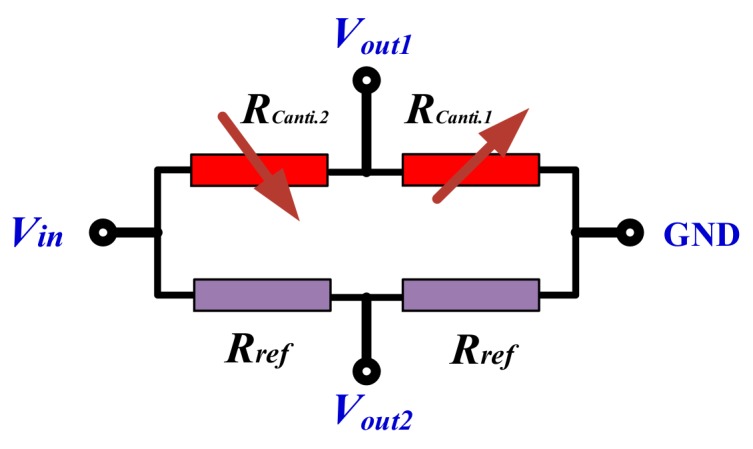
Schematic of the half-bridge circuit for X-axis acceleration sensing.

**Figure 6 micromachines-10-00473-f006:**
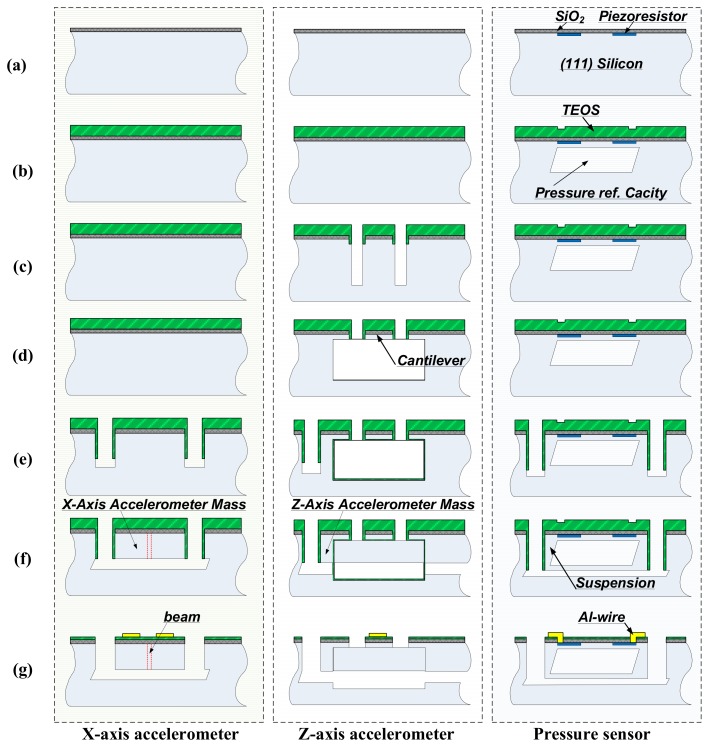
Cross-sectional process steps in ordinally single-polished (111) wafer with the cross-section cut along the broken line of D-D´ in Figure 1.

**Figure 7 micromachines-10-00473-f007:**
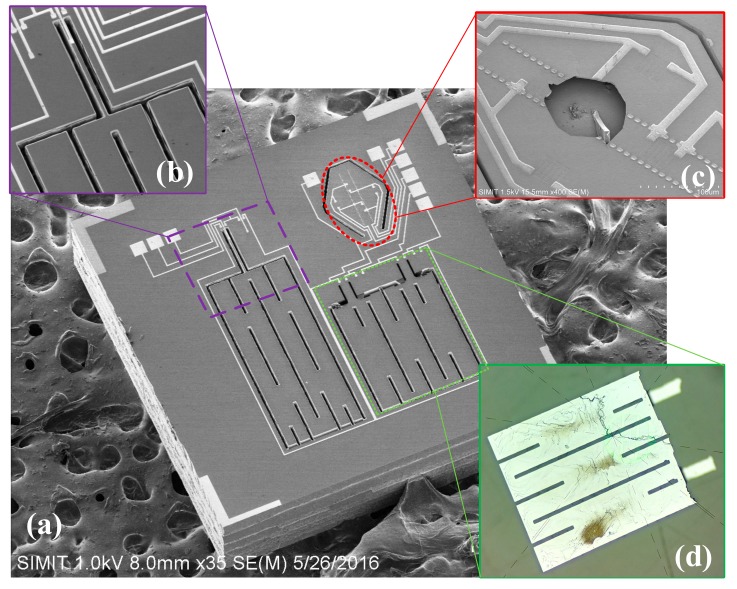
(**a**) SEM image of the fabricated sensor. (**b**) Close-up view of the X-axis accelerometer. (**c**) Close-up view of the pressure sensor with manually broken pressure sensing diaphragm. (**d**) Backside optical microscope image showing the mass of the Z-axis accelerometer.

**Figure 8 micromachines-10-00473-f008:**
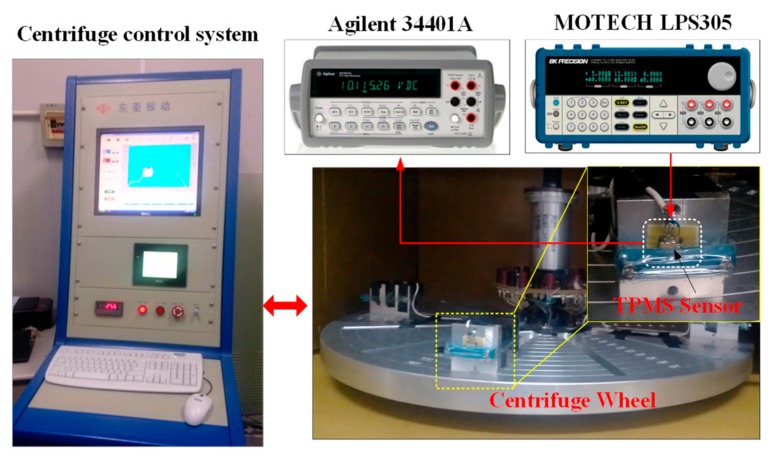
Schematic of the measurement setup for accelerometer.

**Figure 9 micromachines-10-00473-f009:**
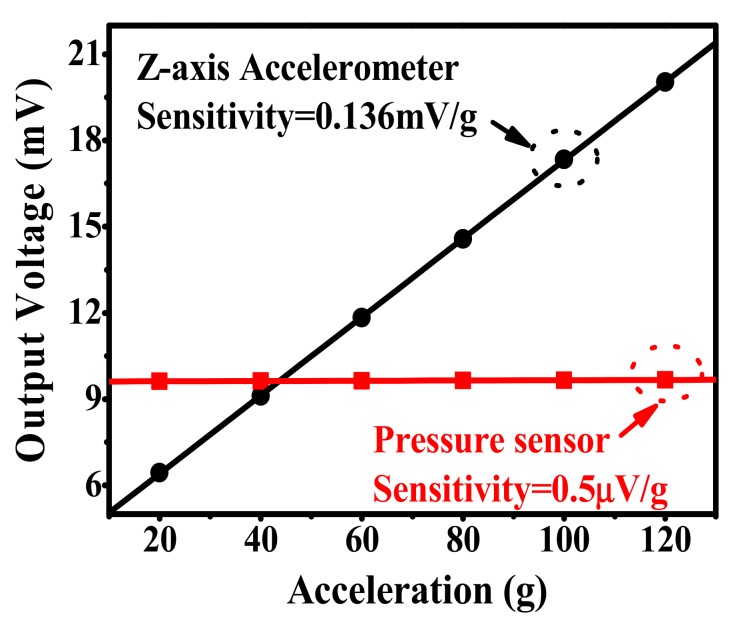
Z-axis accelerometer linear output and the depressed pressure-sensor output induced by the Z-axis acceleration.

**Figure 10 micromachines-10-00473-f010:**
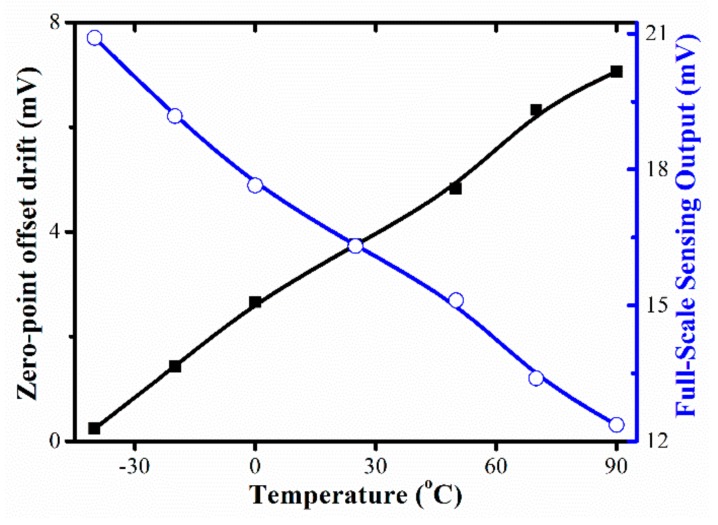
Tested zero-point offset and full-scale output of Z-axis accelerometer in terms of temperature change.

**Figure 11 micromachines-10-00473-f011:**
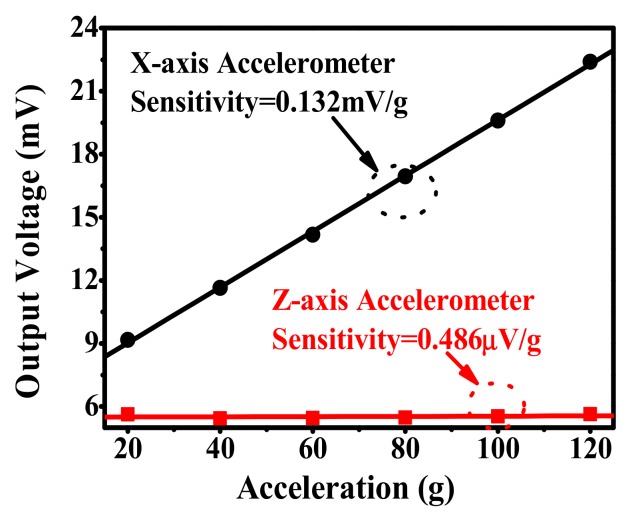
X-axis accelerometer linear output and the eliminated cross-sensitivity of the Z-axis accelerometer, which is caused by the x-axis acceleration.

**Figure 12 micromachines-10-00473-f012:**
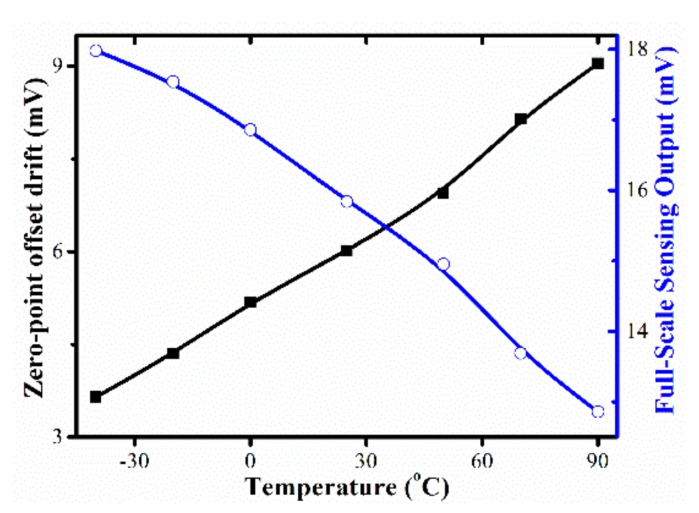
Zero-point offset and sensitivity of the X-axis accelerometer as functions of temperatures.

**Figure 13 micromachines-10-00473-f013:**
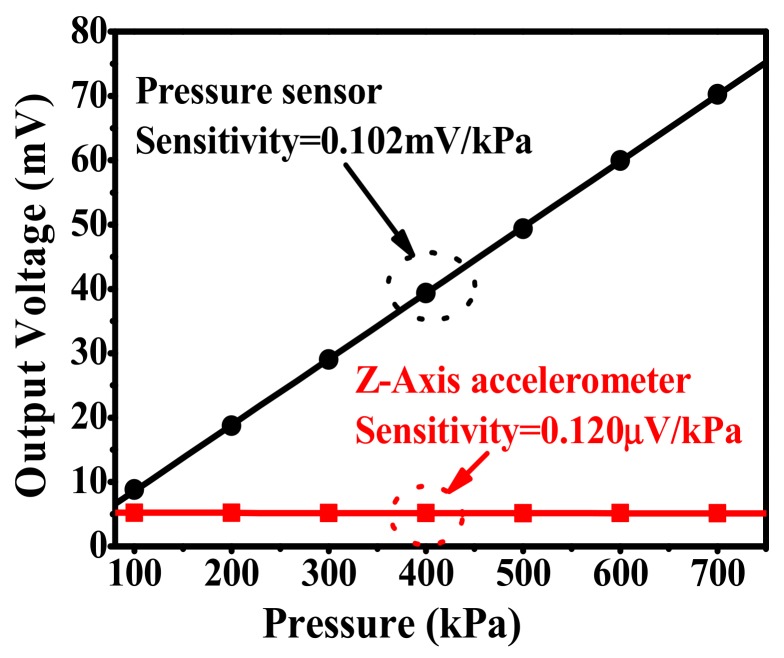
Pressure-sensor linear output and the pressure-induced negligible crosstalk in the Z-axis accelerometer.

**Figure 14 micromachines-10-00473-f014:**
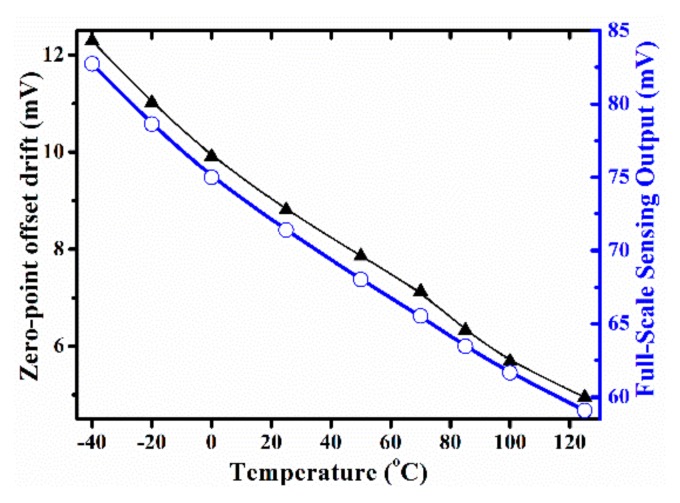
Zero-point offset and full-scale output of the pressure sensor versus temperature.

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
