# Peer review of "On-Chip Integration of Pressure Plus 2-Axis (X/Z) Acceleration Composite TPMS Sensors with a Single-Sided Bulk-Micromachining Technique"

_micromachines, 2019, doi:10.3390/mi10070473_

Round 1

Reviewer 1 Report

Reviewer report

The manuscript reports the design, fabrication and characterization of a composite tire-pressure monitoring system (TPMS) sensor. The topic is of interest, since it has practical implications. However, to increase the relevance of the study, the introduction should include a more detailed state of the art on TPMS with appropriate references. Furthermore, previous published papers by the Author on the proposed sensor should be included among the references, such as the conference paper entitled ‘‘Pressure +X/Z two-axis acceleration composite sensors monolithically integrated in non-soi wafer for upgraded production of TPMS (Tire Pressure Monitoring Systems)’’ (authors: Jiachou Wang, Zao Ni, Jian Zhou, Xuejun Shi, Tiejun Wan and Xinxin Li), where are also reported some figures analogous to those of the actual manuscript. If possible, the figures should be updated. Some detailed comments are reported here below.

Detailed comments

Lines 22-25: The abstract should give a concise overview of the work. The information given here are very detailed and should be, preferably, inserted in the paragraph of conclusion, summarizing in the abstract qualitatively the main findings.

Line 40: ‘’..in the 4S shops’’. Please explain the meaning this expression.

Line 47: ‘’ TPMS sensors have been only consisted’’... Please check the grammar of the sentence.

Line 54: A capital letter is missing. Please check.

Lines 56-58: The sentence is not clear. Please, consider rewriting it or split in more sentences.

Line 92: The final part of the sentence is missing. Please check.

Line 94: The numbers in the horizontal and vertical axes of the diagrams reported in figure 3 are difficult to read and their resolution should be improved.

Lines 95-96: The caption of figure 3 refers to ‘’line a-a’’’ and to ‘line b-b’’’. If these are those depicted in figure 1 it should be stated in the caption and they should be recalled with capital letters (A-A’ and B-B’).

Lines 108-109: In this sentence a word seem missed. Please check.

Line 155: Cross-section D-D’ in figure 1 is barely visible. Please, try to improve it.

Line 165: Please consider the deletion of the expression ‘’Note:..’’, since it is not usual in a main text.

Line 188: ‘’Figure 9(b) shows..’’. It seems that the description is referred to figure 7(b). Please check.

Lines 230-231: The sentence is not clear; please check.

Author Response

Below please find our point-to-point answer and the revision description.

1. Lines 22-25: The abstract should give a concise overview of the work. The information given here are very detailed and should be, preferably, inserted in the paragraph of conclusion, summarizing in the abstract qualitatively the main findings.

Answer: Thank you! The abstract has been re-written to clear state the main findings. The information, Lines 22-25 in original manuscript, have been inserted in the paragraph of conclusion.

2. Line 40: “..in the 4S shops’’. Please explain the meaning this expression.

Answer:4S is the abbreviation of Automobile Sales Service shop.

3. Line 47: “TPMS sensors have been only consisted’’... Please check the grammar of the sentence.

Answer: Thank you for pointing out this! We have corrected the grammar of the sentence as follows: “TPMS sensors have consisted of only a pressure sensor and a Z-axis accelerometer”

4. Line 54: A capital letter is missing. Please check.

Answer: Thanks. I have corrected it.

5. Lines 56-58: The sentence is not clear. Please, consider rewriting it or split in more sentences.

Answer: Thanks. The sentence has been re-written as follows: “More important that unlike the integration of X- and Y-axis accelerometers laid in-plane, integration of vertical Z- and horizontal X-axis accelerometers is quite challenging.”

6. Line 92: The final part of the sentence is missing. Please check.

Answer: Sorry,  the incomplete sentence in Line 92 in original manuscript should be deleted.

7. Line 94: The numbers in the horizontal and vertical axes of the diagrams reported in figure 3 are difficult to read and their resolution should be improved.

Answer: Thanks! Figure 3 have been redrawn.

8. Lines 95-96: The caption of figure 3 refers to ‘’line a-a’’’ and to ‘line b-b’’’. If these are those depicted in figure 1 it should be stated in the caption and they should be recalled with capital letters (A-A’ and B-B’).

Answer: The “line A-A’” and “line B-B’” are depicted in figure 2. It has been stated in the caption of figure 3.

9. Lines 108-109: In this sentence a word seem missed. Please check.

Answer: Thank you for pointing out it. The missing word is add in the revised manuscript.

10.Line 155: Cross-section D-D’ in figure 1 is barely visible. Please, try to improve it.

Answer: Thank you for pointing out it! Cross-section D-D’ in figure 1 has been modified.

11. Line 165: Please consider the deletion of the expression ‘’Note:..’’, since it is not usual in a main text.

Answer: Thank you! I have deleted the expression.

12. Line 188: “Figure 9(b) shows..’’. It seems that the description is referred to figure 7(b). Please check.

Answer: Thank you for your pointing out it. I have corrected it.

13. Lines 230-231: The sentence is not clear; please check.

Answer: The sentence has been corrected as “According to equation (10), the thinner the cantilever the higher the sensitivity.”.

Other revision: Ph.D. Fang Song helped to revise the paper. Now we all agree to add Fang Song as the second author of the paper. Now we have added her in the author list.

Reviewer 2 Report

1. In figure 1, some of the words are unclear due to the overlapping. In addition, some of the important components on device are too small to identify. This figure should be more clearly drawn.

2. In line 92, a part of the text may be missing.

3. In figure 6, the authors should clearly that each column indicates the components of the sensor structure.

4. Direction of acceleration

In figure 9 and 11: To characterize the fabricated X-/Z-axis accelerometer, the acceleration was given to the device only in positive direction, i.e. one-way direction. I think the authors should show measurement results of the sensors in positive/negative directions.

5. Application possibility of 3-axis sensor

The authors should discuss about an application possibility of 3-axis sensor based on their proposed sensor.

Author Response

Below please find our point-to-point answer and the revision description.

1. In figure 1, some of the words are unclear due to the overlapping. In addition, some of the important components on device are too small to identify. This figure should be more clearly drawn.

Answer: Thank you for your good suggestion! The figure 1 have been clearly redrawn.

2. In line 92, a part of the text may be missing.

Answer: Thank you for your pointing out this! the incomplete sentence in Line 92 in original manuscript should be deleted.

3. In figure 6, the authors should clearly that each column indicates the components of the sensor structure.

Answer: Thank you for your advice! We have indicated the components of the sensor structure in each column in figure 6.

4. Direction of acceleration

In figure 9 and 11: To characterize the fabricated X-/Z-axis accelerometer, the acceleration was given to the device only in positive direction, i.e. one-way direction. I think the authors should show measurement results of the sensors in positive/negative directions.

Answer: Since the acceleration measure-range of the X-axis accelerometer and the Z-axis accelerometer is 0g-120g, and the wheel-rotation-induced centrifugal acceleration is only in one-way direction shown in figure 8, we have only tested the acceleration in positive direction in the original manuscript.

5. Application possibility of 3-axis sensor

The authors should discuss about an application possibility of 3-axis sensor based on their proposed sensor.

Answer: The application possibility of 3-axis sensor have been discussed in Line 32-40 in revision.

Other revision: Ph.D. Fang Song helped to revise the paper. Now we all agree to add Fang Song as the second author of the paper. Now we have added her in the author list.